# Psychological Flexibility Is Associated with Parental Stress in Relatives of People with Intellectual Disabilities

**DOI:** 10.3390/ijerph19106118

**Published:** 2022-05-18

**Authors:** David Lobato, Francisco Montesinos, Juan M. Flujas-Contreras

**Affiliations:** 1Department of Psychology, Universidad Europea de Madrid, 28670 Madrid, Spain; francisco.montesinos@universidadeuropea.es (F.M.); jfc397@ual.es (J.M.F.-C.); 2Instituto ACT, 28036 Madrid, Spain; 3Department of Psychology, Universidad de Almería, 04120 Almeria, Spain; 4Centro de Investigación en Salud, Universidad de Almería (CEINSA/UAL), 04120 Almeria, Spain

**Keywords:** parenting, psychological flexibility, disability, contextual therapies, acceptance and commitment therapy, ACT, parental stress

## Abstract

The objective of the study was to examine the relationship between psychological flexibility, perceived stress, and psychological heath in relatives of people with a disability diagnosis. 151 relatives completed an online questionnaire that included 6-PAQ (parental psychological flexibility), PSS (perceived stress), GHQ-12 (psychological health) and WBSI (suppression of unwanted thoughts). The results showed significant relationships between the four measured variables. A bimodal distribution was observed in the variables related to psychological flexibility. The multiple regression showed that difficulties in self as context, committed actions and defusion explain a high percentage of the variance of parental stress and general psychological health. The study provides new evidence to consider psychological flexibility as a mediating variable in psychological well-being. The presented data served as the basis for the development of an ACT-based intervention protocol and the implementation of a clinical trial for relatives of children with disabilities.

## 1. Introduction

The intellectual disability is a specific type of disability that is characterized by limitations and difficulties in intellectual functioning and adaptive behavior, manifesting itself in social skills, practices, and conceptual competencies [1]. The health of the caregivers of people with disabilities is affected by the attention that they have to provide. Multiples studies consistently show a significant higher risk of suffering psychological stress in parents of children with disabilities [2,3]. The high demands of care are one of the main sources of stress. Depending on the type of disability and its etiology, these people might require specific and individualized attention, which for the parents implies an additional responsibility [4]. Behavioural problems, an important factor regarding perceived stress in parents, are added to the demand of care [5]. Likewise, the stigma that surrounds people with disabilities has been identified as an added source of stress, which can lead to changes in the behavior of parents [6], showing harsh and negative behavior patterns and negative perceptions towards children [7]. Furthermore, the time involved on the demands associated with care task might force a reorganization of priorities, affecting the professional development of caregivers, leading to financial pressure and exacerbating perceived stress and other difficulties [8]. The mentioned difficulties imply a compromise of their mental and physical health and they can damage the relationship with the child, what might lead to negligent parenting styles that could effect on the children’s levels of development and global functioning [9]. In addition, the health crisis of COVID-19 is increasing stress levels in the general population [10], and that stress is multiplied in parents of children with disabilities [11].

In the recent decades, a new generation of psychological interventions based on acceptance, psychological flexibility (PF) and mindfulness are showing effectiveness in numerous psychological settings [12]. Among these so-called contextual or third wave behavior therapies, Acceptance and Commitment Therapy (ACT) is one of the most relevant [13] and emphasizes the role of psychological inflexibility (PI) in psychological problems. Literature about the topic is extensive and numerous studies have found a significant relationship between high levels of PI and a wide variety of psychological problems [14]. ACT is oriented towards promoting psychological flexibility (PF) which is defined as a set of behavioural skills that allow to get in touch with private events with aversive functions in a full and conscious way, while orienting behavior towards values [13]. That is, acting towards values regardless of the emotion of cognition that may arise. Research indicates that the interventions that promote parental psychological flexibility reduce stress, improve overall health and leads to interactions connected to what is important to parents or caregivers [15,16]. These programs usually use as an indicator of effectiveness the tendency to suppress unpleasant thoughts, which seems sensitive to the application of the ACT intervention, since it has been found that parents tend to reduce the frequency of avoidant behaviors of unpleasant private events (thoughts and emotions) after ACT treatment [17]. The literature suggests the existence of two trends in families after understanding the life limitations of their children with disabilities: while a part of the parents report higher levels of happiness, maturity, self-realization and well-being [18], the other part reports higher levels of discomfort, perceived stress, as well as a tendency to suffer from other psychological problems.

The main objective of the present study is to analyse the relationship between the following variables: (1) PF, (2) perceived stress levels, (3) suppression of unwanted private events and (4) psychological health in relatives of people with disabilities in the context of COVID-19. In particular, it was intended to study psychological flexibility as a predictor variable of stress and health, with the purpose of creating subsequent programs that could promote the development of psychological flexibility in this population in order to improve their well-being.

## 2. Materials and Methods

### 2.1. Design

In order to explore the relationship between the mentioned variables, a correlational ex post facto retrospective design was used. Four questionnaires were selected to evaluate the mentioned variables in a sample of mothers and fathers of children with disability.

### 2.2. Participants

The inclusion criteria were: (1) being over 18 years old, (2) being the mother, father of main caregiver of a person with a diagnosis of intellectual disability for, at least, six months and (3) an adequate level of understanding of Spanish and the ability to understand informed consent.

151 family members participated; the majority (83.44%) were women with a son or daughter (child or adult) with intellectual disability. The average age of the relatives was 50.83 years (*SD* = 9.64). They had an average of 1.87 (*SD* = 0.86) children per family and the age of the children/adults diagnosed with intellectual disability was 19.79 years (*SD* = 9.11). 71.7% of families had other children than the diagnosed one. 13.37% of the children/adults diagnosed were adopted or fostered. 46.9% of daughters and sons had a diagnosis of mild intellectual disability in the absence of another comorbid difficulty, while 53.1% also had neurodevelopmental or associated medical problems. 80.13% of the participants were married, and 58.27% had completed university studies. The majority of the participants were Spanish (86.75%) and lived in Madrid (76.86%).

### 2.3. Measures

*Perceived Stress Scale* (*PSS*) [19]. The adaptation from [20] were used. It is a widely used measure to evaluate the degree of perceived control over life circumstances. In the study on psychometric properties, Cronbach’s alpha values of 0.727, 0.826 and 0.868 were obtained. It is a one-dimensional scale with 14 items that are answered on a Likert-type scale in a range from 0 (Never) to 5 (Very often). The direct scores range from 0 to 56, higher scores indicate higher perceived stress.

*Parental Acceptance Questionnaire* (*6-PAQ*) [21]. In order to evaluate parental PF the Spanish version of 6-PAQ was used [22]. It’s a 16-item questionnaire on a Likert-type scale with four answer options in a range from 1 (Strongly Disagree) to 4 (Strongly Agree) that evaluates six processes related to PF (being present, values, committed action, self as context, cognitive defusion and acceptance) and tree flexible response styles (opened, centred and committed). The scores vary from 16 to 64, the higher the score, the higher IF levels [22]. In studies on psychometric properties of the scale Cronbach’s alpha scores of 0.81 and McDonald’s omega of 0.86 were reported.

*White Bear Suppression Inventory* (*WBSI*) [23]. The Spanish validation of WBSI [24] was used to evaluate the tendency to suppress unwanted thoughts. It is a Likert scale of 15 items with five response options in a range from 1 (Completely Disagree) to 5 (Completely Agree). Scores range from 15 to 75. Higher scores indicate stronger thought suppression tendency. Regarding the internal consistency of the scale, Cronbach’s alpha values of 0.89 were reported in the general scale and values of 0.87 and 0.80 for its subscales.

*Psychological health Questionnaire* (*GHQ-12*) [25]. It is a widely used measure for the assessment of psychological health. It contains 12 items and its consistency measured with Cronbach’s alpha is 0.85. Higher scores indicate lower levels of psychological well-being. The Spanish validated version was used showing a Cronbach alpha of 0.76 [26].

### 2.4. Procedure

The study was authorized by the European University Research Ethics Committee. A convenience sample was recruited between August and October 2020 through associations, foundations, educational centres and social media networks. The form was anonymously self-administered and completed through the Internet. Every participant accepted the informed consent. Confidentiality and anonymity were guaranteed.

### 2.5. Statistical Analysis

In order to analyse the data descriptive analyses, comparison of means, correlation and predictive analysis through Student’s T, Pearson’s correlation coefficient and multiple lineal regression of successive steps were implemented using the statistical analysis program JASP (0.14 version).

## 3. Results

### 3.1. Descriptive Analysis

Table 1 shows the means and standard deviations obtained in each of the scales, as well as in their subscales.

Regarding the PI, a mean score of 36.6 (*SD* = 13.66) was obtained. The scores observed in the assessed sample were distributed in a range of 46 values, varying from 16 to 62. Therefore, the evaluated sample showed PI at every possible value. Figure 1 shows the distribution of scores obtained on the scale and on each of its subscales included in the concept of PF. A bimodal distribution was observed in relation to the overall score, as well as in the subscales, so data was grouped in two differentiated subsets.

In relation to perceived stress, an average score of 29.49 (*SD* = 12.49) was reported including values that vary from 7 to 50, the most frequent scores were 10 and 45. Regarding psychological health, a mean score of 19.49 (*SD* = 10.25) was obtained involving values ranging from 1 to 36, the most frequent scores were 10 and 30. With regard to unwanted thoughts suppression a mean score of 48.21 (*SD* = 16.51) was reported with values in a range from 18 to 74 in which a part of the subjects are grouped in lower scores (between 20 and 30), while a greater number of participants are grouped in high scores, around 60 points. Figure 2 shows the distribution of direct scores obtained in these questionnaires; a bimodal distribution was again observed.

### 3.2. Analysis of the Scores of Participants That Reported Extreme Levels of PI

Considering the general tendency to bimodal grouping of the scores, the segmentation of the sample into two groups was considered. Since the main variable analysed in the study was PI, the participants’ scores who reported a low PI (group 1) and the participants’ scores who reported high PI (group 2) were examined. The sample was segmented based on the PI cut-off values on percentile 33 and 67 respectively and two extreme groups were contemplated. Group 1 included 58 participants. A mean of 23.63 (*SD* = 2.96) was obtained with a maximum score of 27. Group 2 included 52 participants; the mean score was 53.67 (*SD* = 5.51) and the minimum score was 41. Student’s *t*-test was performed to compare the mean scores for each of the variables evaluated including each scale and subscale. The difference between the means of each group is statistically significant (*p* < 0.001) and a large effect size was reported (see Table 2).

As it is observed in Figure 3, the participants from group 1 (low PI) who scored low on the 6-PAQ also scored low in all the subscales of the scale and showed a low tendency to supress unwanted thought and a high perception of their psychological health.

Regarding the relationships between the measured variables in group 1 (low PI), as it is shown in Table 3, a moderate and statistically significant correlation is found between thought suppression and perceived stress, as well as a high and statistically significant correlation between psychological health and perceived stress. A moderate and statistically significant correlation was also found between psychological health and thought suppression.

On the contrary, a completely different pattern was found in group 2 (high PI). These participants scored higher on every 6-PAQ scale, perceived stress, suppression and psychological health. Figure 4 shows the distributions of the scores of this group.

Concerning the relationships between variables in group two (see Table 4), moderate and statistically significant correlations were found between perceived stress and PI and between perceived stress and thought suppression. Regarding PI and thought suppression, a low and significant correlation was found. A high and statistically significant correlation was found between PI and psychological health, as well as it was found between psychological health and perceived stress. Lastly, a moderate and statistically significant correlation was found between perceived health and thought suppression.

### 3.3. Correlation Analysis and Lineal Regression of the Complete Group

The results suggest the existence of a significant direct relationship between perceived stress, PI, tendency to suppress unwanted thoughts and psychological health, positive correlations of high magnitude and statistically significant between the total scores of the four scales were found. Likewise, statistically significant correlations were found between the scores obtained in the subscales and the global scales of the 6-PAQ and WBSI questionnaires (see Table 5).

Finally, in the multiple regression analysis (Table 6) in which parental stress is the independent variable, a fourth statistically significant model was found (F = 117.78; *p* < 0.001) in which difficulties in self as context (β = 2.047; *t* = 5.56; *p* < 0.001), committed actions (β = 1.765; *t* = 4.14; *p* < 0.001) and defusion (β = 0.772; *t* = 2.18; *p* < 0.001) explain 70.6% of the variance of parental stress (R^2^ Ajust. = 0.700). Similarly, a model in which difficulties in self as context (β = 1.121; *t* = 3.28; *p* = 0.001), committed actions (β = 1.606; *t* = 4.06; *p* < 0.001) and defusion (β = 0.901; *t* = 2.75; *p* = 0.007) explain 62.6% of the variance of psychological health (R^2^ Ajust. = 0.618) was found (F = 81.95; *p* < 0.001) Finally, a statistically significant model (F = 69.58; *p* < 0.001) in which the variance of the suppression of thoughts is explained by 48.5% (R^2^ Ajust. = 0.478), by the difficulties in the self as context (β = 2.392; *t* = 4.27; *p* < 0.001) and by acceptance (β = 2.039; *t* = 3.315; *p* = 0.001) was established.

## 4. Discussion

The objective of the project was to study the existing relationships between PI, perceived stress, thought suppression and psychological health in relatives that look after people with intellectual disabilities. The study aims to enlarge the knowledge about these variables and contribute about the detrimental effects of PI on health while parenting and taking care of children with disabilities.

Regarding the results, the distributions of the obtained scores at the scales and subscales are mostly bimodal. Therefore, the majority of participants are situated on the extremes, showing low PI levels, low thought suppression, good psychological health and low perceived stress levels or just the opposite pattern. This difference could account for the disparities reported in previous studies [18] and might have its origin in the differential impact of the disability on the family construction, their coping strategies, the search for professional support, the degree of the child’s disability or the socioeconomic status among other variables. Likewise, the fact that the parents’ age is high might have allowed the consolidation of these two differentiated patterns.

The segmentation of the sample allows to identify the clear differences depending on the PI level of the participants. Thereby, the relatives that reported lower levels of PI, also displayed low levels of perceived stress, low levels of thought suppression tendency and high levels of psychological health perception. In the low IP group, according to the results of the correlational analysis it can be inferred that they are more focussed on the present, they show a high tendency to emotional acceptance together with a high ability to practice defusion from private events. Moreover, they have clearer values, they display actions oriented to their values and they have a more developed sense of their “self as context” as a consistent perspective from which to observe and accept their changing private experiences. On the contrary, the group that reported higher levels of PI obtained high scores in perceived stress and tendency to suppress thoughts and low psychological health perception. These results are in line with previous literature on PI, which found associations between PI and psychological problems such as stress and anxiety. The data obtained in this study can account for the disparity found in the two lines of scientific research which were previously mentioned. On the one hand, the relatives of people with disabilities are more vulnerable and prone to suffer from psychological problems [9], and on the other hand, they show a high capacity for growth in the face of adversity derived from disability [18].

The results of the correlations and multiple linear regression performed support a strong positive relationship between PI and perceived stress. It has been found that PI, specifically the difficulties in the self as context, committed actions and cognitive fusion, can be predictive variables of stress and psychological health in relatives of children with disabilities, in accordance with the scientific literature. These results seem to be consistent with the COVID-19 pandemic context in families without children with disability [27]. There is scientific evidence that links sustained stress over the time with health problems such as anxiety and depression [28] coronary heart diseases [29] or sleeping problems [30]. The results obtained in the present study can be added to this evidence since it offers data about the high correlation that exist between PI and its impact on psychological health and perceived stress. A positive and strong correlation is observed between PI and general tendency to suppress unwanted thoughts. In the regression analysis it is pointed out that acceptance and “self as context” difficulties can predict thought suppression on the relatives. The correlation shown between both variables was expected and it coincides with the previous literature in the context of disability and developmental disorders [31] and this makes sense considering that one of the components of the PF model is the continuum between experiential acceptance and avoidance. Therefore, the more inflexible a person is, the stronger the effort that he or she will make to supress or avoid his own unpleasant thoughts and experiences. Consistently with the two previous analyses, the observed relationship between perceived stress and thought suppression is high, positive, and statistically significant. This could be explained by the evidence that show that control, modification and thought suppression can have counterproductive effects [32] and this tendency could be contributing to the greater stress perceived in the analysed sample. Since the PF model includes components that try to promote the acceptance of unpleasant feelings and emotions, and a non-judgmental and deliberately open approach to them, it is expected that PF training, with emphasis on the key processes detected (defusion, committed actions, and self as context) translates into a lower tendency to suppress unpleasant thoughts and emotions, and therefore in a reduction of perceived stress. Moreover, it is also expected that PF training could help parents to reconnect with their life project and specifically with the type of parents that they want to be in spite of the presence of emotional pain. The benefits that could be obtained would not be limited caregivers health and stress but could have a positive impact on parenting styles or on the relationship with their children and, therefore, on their levels of development, wellbeing and functionality [33].

This study has several limitations. Firstly, the sample was incidental and obtained contacting associations and participants through social media, so our sample cannot be considered a representative one of the whole population of families with intellectual disabilities. There are also some variables that have not been controlled and could bias the results, as it is mentioned in previous literature, such as gender (predominantly female), age (mostly high) of both parents and children, educational level, number of children, or the type of link with associations, that could influence the impact of the disability in the family system. Other limitations might be related to the online format of the questionnaire (which difficulties the access for a certain sector of the population).

## 5. Conclusions

The obtained data in the correlational study have provided new evidence about the relevance of the PI when intervening with parents of children or adults with disability. It has also contributed to the justification of a subsequent design and study of a program directed to promote PF in this specific population. This study could aim at evaluating whether the intervention would have an impact on perceived stress, thought suppression and psychological health, as well as on the interactions between family members and children with disabilities, as shown in scientific literature.

## Figures and Tables

**Figure 1 ijerph-19-06118-f001:**
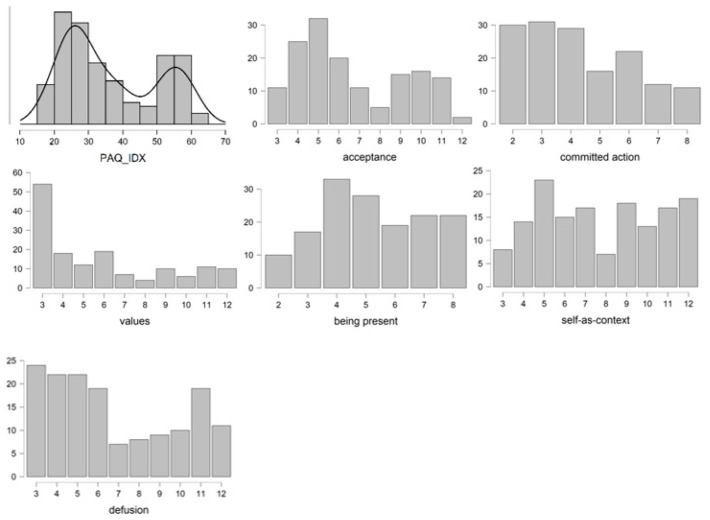
Distribution of the scores of 6-PAQ and its subscales. *Note:* PAQ_IDX: 6-PAQ total score.

**Figure 2 ijerph-19-06118-f002:**
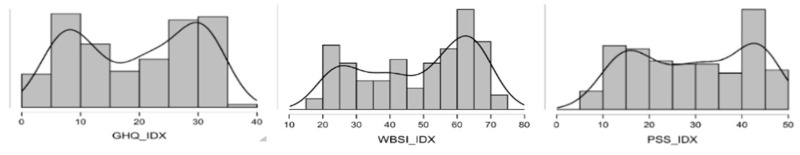
Distribution of the scores of 6-PAQ and its subscales. *Note:* GHQ_IDX: GHQ total score; PSS_IDX: PSS total score; WBSI_IDX: WBSI total score.

**Figure 3 ijerph-19-06118-f003:**
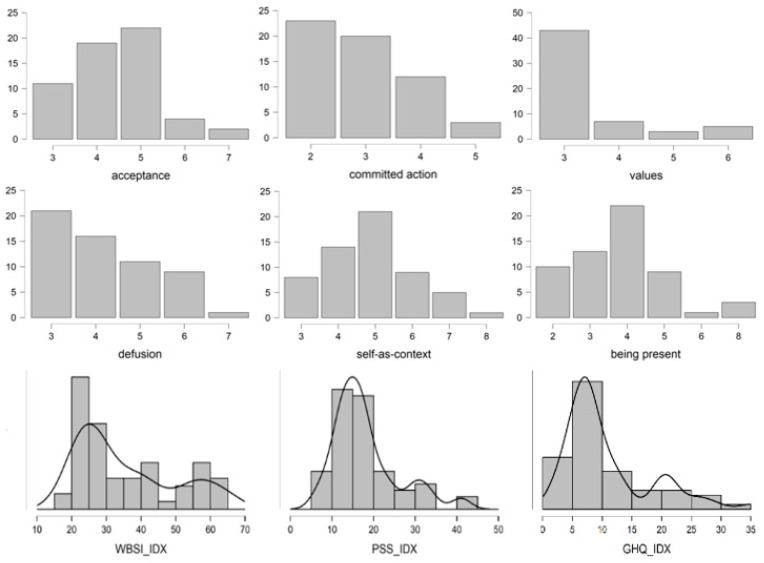
Distribution of the total scores of WBSI, PSS, GHQ and the subscales of 6-PAQ for group 1 (low PI).

**Figure 4 ijerph-19-06118-f004:**
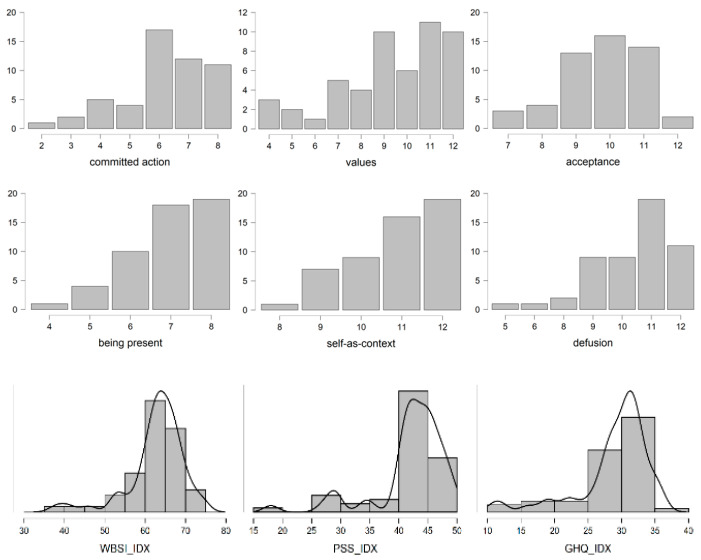
Distribution of the scores of WBSI, PSS, GHQ and the subscales of 6-PAQ for group 2 (high PI).

**Table 1 ijerph-19-06118-t001:** Scores obtained on the scales and subscales for the whole sample.

Variable	M	*SD*
6-PAQ Psychological Inflexibility-Total	36.6	13.6
6-PAQ Acceptance	6.6	2.5
6-PAQ Committed action	4.3	1.8
6-PAQ Values	5.8	3
6-PAQ Being present	5.2	1.8
6-PAQ The self	7.7	2.8
6-PAQ Defusion	6.7	3
PSS Perceived Stress	29.4	12.4
GHQ-12 Psychological health	19.4	10.2
WBSI Thought suppression	48.2	6.5

*Note:* M = mean; *SD* = Standard Deviation.

**Table 2 ijerph-19-06118-t002:** Differences between groups based on levels of psychological flexibility.

Variable	Group 1 (*n* = 58)	Group 2 (*n* = 52)		
	M (*SD*)	M (*SD*)	*t*	*p*
6-PAQ Psychological Inflexibility-Total	23.6 (2.9)	53.6 (5.5)	−36.091	<0.001
6-PAQ Acceptance	4.4 (0.9)	9.7 (1.2)	−25.336	<0.001
6-PAQ Committed action	2,9 (0.9)	6.1 (1.4)	−14.164	<0.001
6-PAQ Values	3.4 (0.9)	9.3 (2.3)	−17.719	<0.001
6-PAQ Being present	3,8 (1.9)	9.9 (1)	−13.228	<0.001
6-PAQ The self	4.8 (1.2)	10.8 (1.1)	−26.948	<0.001
6-PAQ Defusion	4.1 (1.1)	10.3 (1.4)	−24.447	<0.001
PSS Perceived Stress	18 (7.8)	42 (6.1)	−17.698	<0.001
GHQ-12 Psychological health	10.7 (7.3)	28.8 (5.4)	−14.578	<0.001
WBSI Thought suppression	35.5 (14.2)	62.7 (7)	−12.493	<0.001

*Note:* M = mean; *SD* = Standard Deviation; *t* = Student’s *t*-test; *p* = significance.

**Table 3 ijerph-19-06118-t003:** Correlations matrix: 6-PAQ, PSS, GHQ and WBSI for group 1 (low PI, *n* = 58).

Variable	1	2	3	4
1. Psychological Inflexibility	-			
2. Perceived Stress	−0.075	-	-	-
3. Psychological Health	0.066	0.801 ***	-	-
4. Thought Suppression	−0.104	0.627 ***	0.482 ***	-

*** *p* < 0.001.

**Table 4 ijerph-19-06118-t004:** Correlations matrix: 6-PAQ, PSS, GHQ and WBSI for group 2 (high PI, *n* = 52).

Variable	1	2	3	4
1. Psychological Inflexibility	-	-	-	-
2. Perceived Stress	0.660 ***	-	-	-
3. Psychological Health	0.794 ***	0.827 ***	-	-
4. Thought Suppression	0.341 *	0.413 **	0.460 ***	-

* *p* < 0.05, ** *p* < 0.01, *** *p* < 0.001.

**Table 5 ijerph-19-06118-t005:** Correlational Matrix: 6-PAQ, PSS, GHQ y WBSI for the whole sample (*n* = 151).

Variable	1	2	3	4	5	6	7	8	9
1. 6 PAQ	-								
2. Acceptance	0.916 *	-							
3. Committed action	0.827 *	0.791 *	-						
4. Values	0.915 *	0.802 *	0.815 *	-					
5. Being present	0.797 *	0.694 *	0.598 *	0.635 *	-				
6. Self-as-context	0.916 *	0.793 *	0.673 *	0.784 *	0.692 *	-			
7. Defusion	0.934 *	0.824 *	0.692 *	0.832 *	0.719 *	0.837 *	-		
8. PSS	0.827 *	0.739 *	0.709 *	0.758 *	0.594 *	0.803 *	0.762 *	-	
9. GHQ	0.783 *	0.681 *	0.688 *	0.720 *	0.588 *	0.733 *	0.731 *	0.919 *	
10. WBSI	0.690 *	0.649 *	0.524 *	0.593 *	0.576 *	0.668 *	0.627 *	0.804 *	0.757 *

* *p* < 0.001.

**Table 6 ijerph-19-06118-t006:** Multiple stepwise regression analysis for parental stress, general health and thoughts suppression and parental psychological flexibility as dependent variable.

	Unstandardized			Standardized	
	β	*Tip. Error*	*Beta*	*t*	*p*
*IV: Parental stress*					
1. Self	2.047	0.368	0.468	5.565	<0.001
2. Actions	1.765	0.426	0.264	4.140	<0.001
3. Defusion	0.772	0.353	0.188	2.183	0.031
*IV: General Health*					
1. Self	1.121	0.341	0.312	3.289	0.001
2. Actions	1.606	0.395	0.293	4.068	<0.001
3. Defusion	0.901	0.327	0.268	2.753	0.007
*IV: Thought supression*					
1. Self	2.392	0.559	0.414	4.276	<0.001
2. Acceptance	2.039	0.615	0.321	3.315	0.001

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
