# Peer review of "Psychological Flexibility Is Associated with Parental Stress in Relatives of People with Intellectual Disabilities"

_ijerph, 2022, doi:10.3390/ijerph19106118_

Round 1

Reviewer 1 Report

Point 1: Currently, in care and support for people with disabilities, there is talk of the importance of the "life project", both of the people with disabilities themselves and of their parents or relatives (being the type of father/mother I want to be ). It could be interesting to refer to the fight against unpleasant thoughts and feelings based on the fact that people who go through the diagnosis of disability in their children find that their life project has faltered and that the relationship with all this can make a difference. Related to the above, justify interventions based on acceptance, psychological flexibility, and mindfulness, on the basis of parents' struggle with difficult thoughts and feelings related to disability and setting them apart from the type of parents what they want to be

Point 2: I wonder if lines 56-57 could positively refer to “less aversive family interactions”. Perhaps talk about "promoting interactions more connected with what is important to one as a father or mother"

Point 3: 62 to 65 I invite a review of this paragraph since the difference between the two trends is not very well understood, with respect to what. These two trends or lines are better explained in the conclusions.

Point 4: Table 1 there is an error in the fourth line of the SD. There is a parenthesis.

Point 5: Regarding the normative data of each measure, it could be useful to identify or point out which scores would mark the differences between high or low values ​​in said variable. Or at least point to the middle value.

Point 6: Figure 2. The title of this figure is repeated with figure 1. It is a misprint.

Point 7: The paragraph between lines 239 and 241 refers to the fact that “relatives who reported lower levels of IP also showed low levels of perceived stress, low levels of tendency to suppress thoughts and high levels of perception of anxiety. psychological health”. However, I miss an interpretation of why in the group with low Psychological Inflexibility, this variable does not correlate with perceived stress, psychological health and with thought suppression, within said group.

Author Response

Response to Reviewer 1 Comments

Point 1: Currently, in care and support for people with disabilities, there is talk of the importance of the "life project", both of the people with disabilities themselves and of their parents or relatives (being the type of father/mother I want to be). It could be interesting to refer to the fight against unpleasant thoughts and feelings based on the fact that people who go through the diagnosis of disability in their children find that their life project has faltered and that the relationship with all this can make a difference. Related to the above, justify interventions based on acceptance, psychological flexibility, and mindfulness, on the basis of parents' struggle with difficult thoughts and feelings related to disability and setting them apart from the type of parents what they want to be.

Response: In line 286 next sentence has been added: “…Moreover, it is also expected that PF training could help parents to reconnect with their life project and specifically with the type of parents that they want to be in spite of the presence of emotional pain”

Point 2: I wonder if lines 56-57 could positively refer to “less aversive family interactions”. Perhaps talk about "promoting interactions more connected with what is important to one as a father or mother":

Response: In line 58-59 next sentence has been added: “…and leads to interactions connected to what is important to parents or caregivers”

Point 3: 62 to 65 I invite a review of this paragraph since the difference between the two trends is not very well understood, with respect to what. These two trends or lines are better explained in the conclusions.

Response: In line 63-64 next sentence has been added: “The literature suggests the existence of two trends in families after understanding the life limitations of their children with disabilities:”

Point 4: Table 1 there is an error in the fourth line of the SD. There is a parenthesis.

Response: Parenthesis deleted.

Point 5: Regarding the normative data of each measure, it could be useful to identify or point out which scores would mark the differences between high or low values ​​in said variable. Or at least point to the middle value:

Response: We think your suggestion is explained in the text: “…The sample was segmented based on the PI cut-off values on percentile 33 and 67 respectively and two extreme groups were contemplated. Group 1 included 58 participants. A mean of 23.63 (SD=2.96) was obtained with a maximum score of 27. Group 2 included 52 participants; the mean score was 53.67 (SD=5.51) and the minimum score was 41”

Point 6: Figure 2. The title of this figure is repeated with figure 1. It is a misprint

Response: Figure 2 tittle changed in line 171: “Differences between groups based on levels of psychological flexibility “

Point 7: The paragraph between lines 239 and 241 refers to the fact that “relatives who reported lower levels of IP also showed low levels of perceived stress, low levels of tendency to suppress thoughts and high levels of perception of anxiety. psychological health”. However, I miss an interpretation of why in the group with low Psychological Inflexibility, this variable does not correlate with perceived stress, psychological health and with thought suppression, within said group

Response: In line 246-247 next sentence had been added: “In the low IP group, according to the results of the correlational analysis it can be inferred that”

Reviewer 2 Report

In the manuscript authors present the useful informtion about Psychological flexibility is associated with parental stress in relatives of people with intellectual disabilities.The study is excellently set up with clear exclusionary factors.The material and methods are very clear, and the results are statistically well presented.Some recent references should be provided before publication

Prior to publication, more structuring and checking of language and grammar is suggested.

Author Response

Response to Reviewer 2 Comments

Point 1: In the manuscript authors present the useful information about Psychological flexibility is associated with parental stress in relatives of people with intellectual disabilities. The study is excellently set up with clear exclusionary factors. The material and methods are very clear, and the results are statistically well presented. Some recent references should be provided before publication. Prior to publication, more structuring and checking of language and grammar is suggested.

Response: Thorough English revision has been done by a native English speaker

Reviewer 3 Report

Dear Authors,

My comments are:

The article looks very interesting, but there are unfortunately many shortcomings in terms of results:

Table 1: display only one decimal digit, there is a bracket in the table, please remove

Table 2: display only one decimal digit for M and SD, Cohen's d values should be positive and are too high, for PI total score I calculated 6.6, perhaps because group 2 PI values do not match in text and table (53.67 vs 52.66), please correct, perhaps “df” is not needed, in heading please add n=58 for group 1 and n=52 for group 2

in some cases you state WBSI Though suppression (in text and tables) please correct to thought suppression

Table 6 for linear regression is missing, please display B (95% CI for B), standard errors and p values, beta and t are not needed, results of all models could or should be presented in one table, linear regression represents key result and should be presented in a transparent manner with all the data needed for easy understanding (not only described in one paragraph)

Also, the last paragraph of the results is described inaccurately, you define parental stress as independent variable and you later state that all predictors explain 70.6% of the variance of parental stress

Please provide accurate understanding of the design of regression models, as already asked before appropriate table presentations would increase understanding of the research.

Please provide description that if I understand correctly parental stress is defined by PSS (perceived stress)?

Please correct R2 Ajust. =0,700 to 0.700, also in some cases p<.001 to p<0.001

Author Response

Response to Reviewer 3 Comments

Point 1: Table 1: display only one decimal digit, there is a bracket in the table, please remove.

Response: Added suggestion: Displayed only one decimal digit.

Point 2: Table 2: display only one decimal digit for M and SD. Cohen's d values should be positive and are too high, for PI total score I calculated 6.6, perhaps because group 2 PI values do not match in text and table (53.67 vs 52.66), please correct, perhaps “df” is not needed, in heading please add n=58 for group 1 and n=52 for group 2

Response: Changes have been applied to table 2: Displayed only one decimal digit. M (SD) corrected, df deleted and n added. The sample is split as an independent variable according to the 6PAQ scores, so the effect size score is high.  This splitting is explained in the method section.

Point 3: In some cases, you state WBSI Though suppression (in text and tables) please correct to thought suppression

Response: Though suppression corrected to thought suppression.

Point 4: Please correct R2 Ajust. =0,700 to 0.700, also in some cases p<.001 to p<0.001

Response: Changes done.

Point 5: Table 6 for linear regression is missing, please display B (95% CI for B), standard errors and p values, beta and t are not needed, results of all models could or should be presented in one table, linear regression represents key result and should be presented in a transparent manner with all the data needed for easy understanding (not only described in one paragraph).  Also, the last paragraph of the results is described inaccurately, you define parental stress as independent variable and you later state that all predictors explain 70.6% of the variance of parental stress .

Please provide accurate understanding of the design of regression models, as already asked before appropriate table presentations would increase understanding of the research. Please provide description that if I understand correctly parental stress is defined by PSS (perceived stress)?

Response: Table 6 added: Linear regression data displayed, hopefully resolving the observations. Yes, PSS assesses parental stress.